# Curli-independent defense against *Bdellovibrio bacteriovorus* in *E. coli*

Ryan Sayegh,[1,2] Hannah E. Ledvina,[1] Aaron T. Whiteley[1]

**ABSTRACT** Predatory bacteria are a group of organisms that use diverse methods to access nutrients and grow by killing prey bacteria. The predator *Bdellovibrio bacteriovorus* is capable of preying on a wide range of gram-negative bacteria by invading the periplasmic space, killing, digesting, and ultimately lysing prey cells. *B. bacteriovorus*, like a phage, replicates at the expense of its host, yet unlike phage defense, there are few characterized mechanisms for bacteria to resist *B. bacteriovorus*. Previously, we discovered that an extracellular amyloid protein called curli protects *Escherichia coli* from *B. bacteriovorus*. Here, we searched for additional modes of *B. bacteriovorus* resistance and identified a strain within the *E. coli* Reference (ECOR) collection, ECOR29, that uses a curli-independent mechanism that requires lipopolysaccharide (LPS)-modifying enzymes for defense. Over 30% of the ECOR collection is resistant to *B. bacteriovorus*. We successfully deleted the gene encoding the major curli subunit in many of these, and only ECOR29 remained resistant. We hypothesized that ECOR29 encoded an alternative resistance mechanism and identified determinants of defense using a forward genetic screen. Our screen revealed critical roles for enzymes that modify LPS, alter the outer membrane, and are homologous to plasmid partitioning systems. Examination of ECOR29 by electron microscopy did not identify overt phenotypes or visible alterations to extracellular structures. We also were unable to identify any secreted factors that impacted *B. bacteriovorus* viability. Our work demonstrates that *E. coli* encode curli-independent mechanisms that restrict *B. bacteriovorus* and expand our understanding of the antipredatory bacteria arm of the bacterial immune system.

**IMPORTANCE** Understanding host-pathogen interactions has the potential to illuminate fundamental aspects of biology. Here, we investigate an atypical host-pathogen system, the interaction between *Escherichia coli* and the predatory bacterium *Bdellovibrio bacteriovorus*. *B. bacteriovorus* has a unique predatory life cycle that requires intimate interactions with the outer membrane, periplasm, peptidoglycan, and inner membrane of prey cells. Accordingly, understanding mechanisms of *B. bacteriovorus* predation and resistance will help us to better understand the gram-negative cell envelope, an ideal target for novel antibacterial compounds. Predatory bacteria are abundant and ubiquitous threats to bacteria in a wide variety of environments. Further findings from experiments in this field will expand our understanding of some of the most basic aspects of the microbial world.

**KEYWORDS** *Bdellovibrio bacteriovorus*, ECOR, curli, BALOs, O-antigen, predatory bacteria

P redatory bacteria are species that kill and consume other microorganisms as part of their predatory life cycles. One such organism is *Bdellovibrio bacteriovorus,* a predator of gram-negative bacteria that replicates intraperiplasmically within prey cells. During predation, *B. bacteriovorus* collides with a prey cell and forms a pore to alter the prey's

Address correspondence to Aaron T. Whiteley, aaron.whiteley@colorado.edu.

The authors declare no conflict of interest.

See the funding table on p. 15.

peptidoglycan and cross the outer membrane. *B. bacteriovorus* next establishes a growth niche along the inner membrane of the prey cell where it grows filamentously until prey nutrients are exhausted. Finally, *B. bacteriovorus* completes the predatory cycle by lysing the prey cell to release progeny (1–5).

*B. bacteriovorus* is capable of preying on a surprisingly large range of gram-negative bacteria, and investigation of the mechanisms of predation has revealed extensive interactions with the outer membrane and remodeling of the host peptidoglycan. Predation starts when *B. bacteriovorus* uses mosaic adhesive trimer proteins to interact with the prey outer membrane (6). Type IV pili are then required to cross the outer membrane (7) and invade the periplasm. *B. bacteriovorus* next secrete peptidoglycan hydrolases that modify prey peptidoglycan and produce the characteristic bdelloplast-rounded prey cell morphology, which prevents superinfection and establishes the predatory niche (8). *B. bacteriovorus* protects its own peptidoglycan using small ankyrin repeat proteins (9). After consuming the prey cell nutrients, the predatory life cycle is completed by initiating prey cell lysis. This liberation is mediated by a lysozyme, DslA, which targets deacetylated peptidoglycan for prey exit (10). Many of the molecular details for predation are poorly understood; however, by advancing our knowledge of how *B. bacteriovorus* hijacks prey cells, we will gain unique insights into fundamental aspects of the bacteria cell envelope.

Although *B. bacteriovorus* appears to indiscriminately prey on gram-negative bacteria, a number of resistance determinants have been documented for prey cells (11). *Vibrio cholerae* cell motility can exert a drag force on predator cells to limit invasion (12). *Acidovorax citrulli* prey cells encode a type II secretion system that can impact susceptibility to *B. bacteriovorus* predation (13). Additionally, *Aquaspirillum sinuosum, Aeromonas salmonicida,* and *Caulobacter crescentus* use paracrystalline protein surface arrays (i.e., S-layers) to restrict *B. bacteriovorus* predation (14). In low nutrient conditions, *Chromobacterium piscinae* can produce sufficient concentrations of cyanide to antagonize *B. bacteriovorus* predation (15). And finally, a recent study reported that K-12 *Escherichia coli* can evolve resistance upon multiple exposures to predators, mediated through mutations in outer membrane proteins or lipopolysaccharide (LPS)-modifying enzymes (16).

Our lab previously searched for novel mechanisms of defense against *B. bacteriovorus* using an approach we termed ExpND (for exploring wild strains for novel defense). First, a diverse collection of *E. coli* strains was screened for resistance to *B. bacteriovorus,* and 27 out of 72 strains were found to be highly resistant (17). A forward genetic screen for mutations that disrupted resistance in one of these strains revealed that curli, proteinaceous extracellular fibers, provided robust protection (17). Curli are composed of the amyloid protein CsgA, which forms fibers anchored to the outer membrane of *E. coli,* and play an important role in biofilm formation (18). Further experiments demonstrated that curli construct a molecular "suit of armor" to protect *E. coli* growing in suspension, not simply bacteria growing in a biofilm, from *B. bacteriovorus*.

Here, we sought to identify additional mechanisms for *B. bacteriovorus* defense by investigating other strains of *E. coli* that were highly resistant. We found that a curli-independent mode of *B. bacteriovorus* defense could only be found for a single strain within the *E. coli* Reference (ECOR) collection, ECOR29. We identified the genetic determinants of ECOR29 resistance and investigated the roles of alternative extracellular structures, secreted bactericidal factors, and ECOR29 mobile genetic elements in restricting predation.

## RESULTS

### Curli-independent *B. bacteriovorus* defense

Resistance against *B. bacteriovorus* is widespread in *E. coli* and can be mediated by the production of curli fibers (17). While curli production is sufficient, it is unclear if there are additional mechanisms *E. coli* can use for defense. We searched for curli-independent defense mechanisms by deleting the major subunit of curli fibers, *csgA*, in a

large collection of resistant *E. coli* and challenging with *B. bacteriovorus*. Previously, we found that 27 strains in the 72 within the *E. coli* Reference collection were resistant to *B. bacteriovorus* HD100. The *csgA* gene could be deleted from 23 of these strains (see Materials and Methods for discussion of technical limitations). An initial analysis of nine strains showed that all relied on curli for defense (17). In this study, we repeated our initial experiments and expanded to include 14 additional curli-deficient strains. Each strain was challenged with *B. bacteriovorus,* and only one strain, ECOR29, remained resistant to predation (Fig. 1a; Fig. S1a). ECOR29 was isolated from a Kangaroo rat in Nevada, USA (19), is most closely related to ECOR32, 33, and 34, and is predicted to have an R1-type core oligosaccharide and an O150:H21 serotype (20).

To confirm that ECOR29 resisted *B. bacteriovorus* independent of curli production and all curli secretion machinery, we constructed a marked deletion of both divergently transcribed operons *csgBAC* and *csgDEFG* (Δcurli). ECOR29Δcurli remained resistant to *B. bacteriovorus*, demonstrating that defense does not require any curli secretion machinery or subunits (Fig. 1b). We then used transmission electron microscopy to visualize the cell surface of ECOR29 in the presence and absence of *csgA*. We compared these images with an *E. coli* strain that uses curli for defense against *B. bacteriovorus* (ECOR14) in the presence and absence of *csgA*. While curli was clearly visible in wild-type ECOR14 and absent in the *csgA* mutant, there were no striking differences between ECOR29, ECOR29

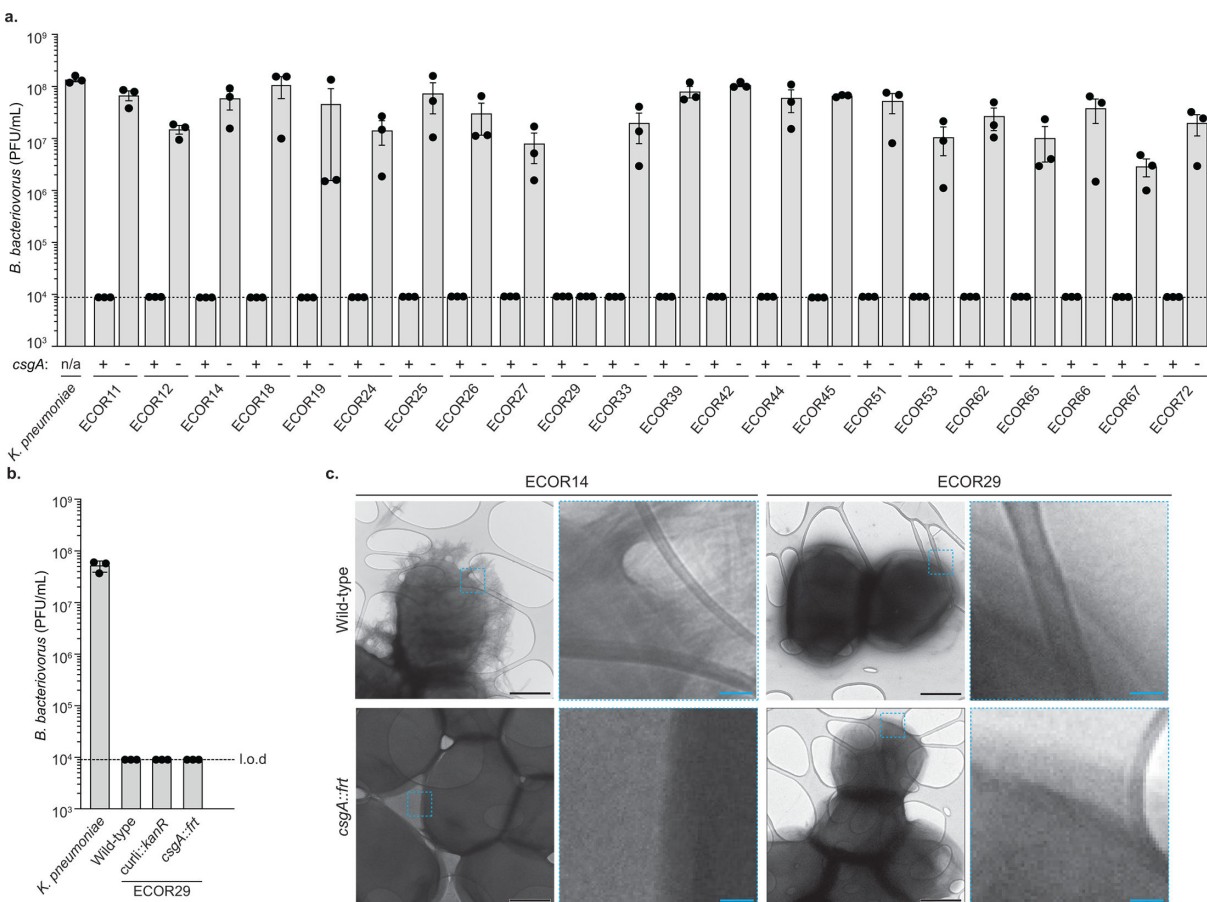

**FIG 1** A curli-independent *B. bacteriovorus* defense mechanism. (a) Efficiency of plating, calculated as plaque forming units (PFU) per milliliter, of *B. bacteriovorus* HD100 when infecting the indicated bacterial strain. The limit of detection (l.o.d.) is indicated by a dotted line. Data are the mean ± standard error of the mean for *n* = 3 biological replicates. (b) Efficiency of plating of *B. bacteriovorus* HD100 when infecting indicated bacterial strain. Data are graphed as in Fig. 1a. (c) Representative transmission electron microscopy images of indicated bacterial strain alongside zoom-in of area depicted by the blue dotted-line box to examine outer membrane features. Black scale bars are 500 nm, and blue scale bars are 50 nm.

*csgA::kanR*, and ECOR14 *csgA::kanR* (Fig. 1c). These findings show that ECOR29 restricts *B. bacteriovorus* predation independent of curli production.

## A forward genetic screen for determinants of defense in ECOR29

We searched for the genetic determinants of ECOR29 defense against *B. bacteriovorus* by transposon-mutagenizing ECOR29 *csgA::frt* and screening mutants for loss of *B. bacteriovorus* resistance. We performed our genetic screen in an ECOR29 *csgA* mutant cured of the kanamycin resistance gene, resulting in an *frt* site replacing the *csgA* locus, to ensure our findings would remain independent of curli. A total of 5,640 transposon mutants were individually challenged with *B. bacteriovorus* in 24-well plates using a double-agar overlay assay to visualize plaque formation. *B. bacteriovorus* did not form a plaque on ECOR29 *csgA::frt* in these conditions but did form plaques on 65 transposon mutants. We repeated our analysis of these 65 mutants in 6-well plates, which allowed for more accurate quantification. Among these, 41 transposon mutants were validated as newly susceptible to *B. bacteriovorus*. When possible, the transposon insertion site was identified for the mutant strains, and the corresponding locus from MG1655 was used to determine annotations and gene names (Table 1; Table S1).

The susceptible transposon mutants fell into two broad categories based on the morphology of the *B. bacteriovorus* plaque formed on that strain. Mutants in group 1 yielded clear plaques, and mutants in group 2 resulted in turbid plaques (Fig. 2a). To uncover the genetic determinants of defense in ECOR29, we focused our attention on the mutants in group 1, which should represent a complete loss of defense. Group 1 contained 23 susceptible mutants representing disruption of 21 unique genes and 1 promoter region (Fig. 2b; Table 1; Table S1). Interestingly, some mutants in group 2 were disrupted in genes found in group 1, such as *mlaA* and *rfe*. However, in group 2 mutants, these genes were disrupted in their predicted promoter regions (*mlaA*) or at the end of the gene (*rfe*). This may indicate that group 2 mutations simply diminish, as opposed to eliminate defense.

## Genetic determinants of ECOR29 defense suggest a role for extracellular modifiers

We validated the results of our forward genetic screen by constructing marked deletions for each of the 21 unique genes disrupted in group 1 mutants. Mutations were introduced into ECOR29 and ECOR29 *csgA::frt* backgrounds. We deleted *lacZ* to construct a negative control strain and showed that knocking out a neutral locus had no effect on *B. bacteriovorus* defense (Fig. S2). This collection of strains was then challenged with *B. bacteriovorus* (Fig. 3). Our analysis revealed that the genes we identified fall into three broad categories: genes that contribute to defense only in the absence of *csgA,* genes that were selectively essential, and genes that contribute to defense independent of genetic background.

Deletion of *mlaA* or *rbsK* resulted in loss of defense in ECOR29 *csgA::frt* but not in ECOR29 (Fig. 3b). The *mlaA* gene encodes an outer membrane lipoprotein responsible for maintaining outer membrane lipid asymmetry (21, 22). Deletion of *mlaA* results in the accumulation of phospholipids in the outer membrane, which causes improper LPS packing and increases membrane permeability (23). The *rbsK* gene encodes a sugar kinase that specifically binds to the α-furanose form of ribose (24, 25). A *rbsK* mutant cannot use D-ribose as a sole carbon source because the RbsK kinase is required to phosphorylate D-ribofuranose early in ribose metabolism (25).

Two additional genes (*ACS228_24475* and *rbsR*) were selectively essential; we were unable to generate deletions of these genes in wild-type ECOR29, but they could be deleted in the *csgA::frt* mutant background. *ACS228_24475* encodes a hypothetical protein that is homologous to the plasmid partitioning gene ParB. In those homologs, ParB binds to specific *parS* DNA sequences and recruits ParA to partition plasmids into daughter cells (26). RbsR is a transcriptional repressor of the ribose operon. D-ribose binds to RbsR to derepress transcription of ribose metabolism genes (25).

**TABLE 1** Group 1 transposon mutants[a]

| TnMut_ID | Disrupted locus | Gene name[b] | MG1655 locus | Description (gene name; predicted function)[b] |
|---|---|---|---|---|
| 3900 | ACS228_04490::TA0368 | *fau* | b2912 | Folinic acid utilization; 5-formyltetrahydrofolate cyclo-ligase which can inhibit various folate-dependent enzymes. Interestingly, the transcription of this gene is induced upon biofilm formation. |
| 2726 | ACS228_19010::TA-096[d] | *ispU* | b0174 | Isoprenoid U; undecaprenyl pyrophosphate synthase that contributes to synthesizing the lipid carrier that localizes carbohydrates to the extracellular space. |
| 0869 | ACS228_03385::TA0410 | *tolC* | b3035 | Colicin-tolerant C; the outer membrane component of the *E. coli* multidrug efflux pump. |
| 4283 | ACS228_07640::TA0837 | *arnD* | b2256 | N/A; predicted to function as a deformylase that removes the formyl group from undecaprenyl phosphate-L-Ara4FN in arabinose metabolism. |
| 4673 | ACS228_19480::TA0684 | *cra* | b0080 | Catabolite repressor activator; dual transcriptional regulator of many genes involved with metabolic pathways which contribute to carbon flow. |
| 2350 | ACS228_24475::TA0336 | N/A[c] | N/A | Predicted ParB family protein; gene encoding a plasmid partitioning protein which works in tandem with ParA to facilitate proper separation of plasmids during replication for vertical transmission. |
| 0805 | ACS228_00410::TA0392 | *waaV*[c] | N/A | Glycosyl transferase family 2[c]; R1-type LPS-specific glycosyl transferase. Mediates the addition of the beta-linked Glc residue on the outer core of the LPS generating the attachment site for O-antigens. |
| 0778 | ACS228_07210::TA0382 | *mlaA* | b2346 | Maintenance of outer membrane lipid asymmetry A; removes glycerophospholipids from the outermembrane and works with outer membrane proteins for the transport of phospholipids. |
| 1377 | ACS228_10665::TA0012 | *lpp* | b1677 | Murein lipoprotein; outer membrane lipoprotein. Tethers the outer membrane to the peptidoglycan layer and dictates the distance between the inner membrane (IM) and outer membrane (OM). |
| 1441 | ACS228_08755::TA0295 | *wzzB* | b2027 | Chain length determinant B; an O-antigen co-polymerase in Wzx/Wzy-dependent pathways. Group 1a polysaccharide co-polymerase that regulates the length of the O-antigen. |
| 3015 | ACS228_22995::TA0914 | *wzzE* | b3785 | Chain length determinant E; part of the Wzx/Wzy-dependent pathway that modulates the length of enterobacterial common antigen, an outer membrane polysaccharide. |
| 2537 | ACS228_08725::TA1015 | N/A[c] | N/A | Predicted O150 O-antigen co-polymerase; works to form chains of polysaccharides that are then secreted to form the O-antigen, which is attached to the LPS outer core. |
| 5140 | ACS228_22690::TA0422 | *rfaH* | b3842 | Transcriptional antiterminator RfaH; regulates LPS, exopolysaccharides, F-pili, and hemolysin expression through modulating RNA polymerase. |
| 2324 | ACS228_22690::TA0435 | *rfaH* | b3842 | Transcriptional antiterminator RfaH; regulates LPS, exopolysaccharides, F-pili, and hemolysin expression through modulating RNA polymerase. |
| 0975 | ACS228_21725::TA0787 | *pgi* | b4025 | Phosphoglucose isomerase; catalyzes interconversion of glucose-6-phosphate and fructose-6-phosphate in glycolysis and gluconeogenesis. |
| 4680 | ACS228_21725::TA1166 | *pgi* | b4025 | Phosphoglucose isomerase; catalyzes interconversion of glucose-6-phosphate and fructose-6-phosphate in glycolysis and gluconeogenesis. |
| 0808 | ACS228_17680::TA0541 | *proC* | b0386 | Pyrroline-5-carboxylate reductase; reduce (S)-1-pyrroline-5-carboxylate to L-proline, a later step in proline synthesis. |
| 5234 | ACS228_23155::TA0416 | *rbsK* | b3752 | Ribose kinase; phosphorylates ribose at O-5a in an ATP-dependent manner which is an early stage in ribose metabolism in the cell. |
| 4464 | ACS228_23150::TA0908 | *rbsR* | b3753 | Ribose repressor; transcription factor that regulates expression of genes that conduct ribose catabolism and transport in *E. coli*. |
| 1159 | ACS228_05370::TA0142 | *rpoS* | b2741 | RNA polymerase sigma factor; alternative sigma factor involved in regulating the stress response (e.g., increasing expression of stress response genes) in response to stressors like outer membrane disruption. |
| 4543 | ACS228_22265::TA0355 | *glpX* | b3925 | Glycerol kinase; type II fructose 1,6-bisphosphatase. Deletion of *glpX* increased *acrAB* expression, which is the other component of the multidrug efflux pump. |
| 5614 | ACS228_00405::TA0547 | N/A[c] | N/A | N/A; a predicted UDP-galactose-(galactosyl) LPS alpha1,2-galactosyltransferase. Involved in galactose synthesis and LPS core biosynthesis. |
| 1658 | ACS228_23000::TA0353 | *rfe* | b3784 | Undecaprenyl phosphate GlcNAc-1-phosphate transferase; catalyzes the transfer of N-acetylglucosamine (GlcNAc)-1-phosphate onto undecaprenyl phosphate leading to the formation of Und-P-P-GlcNAc; the first step in making enterobacterial common antigen. |

[a]Identified transposon mutants were arbitrarily numbered in the order they were collected (TnMut_ID). The transposon insertion site was mapped, and the disrupted locus is presented as the ECOR29 locus tag followed by the location of the transposon insertion site. The transposon is inserted at TA sites within the genome, the numeric values indicating the nucleotides from the 5′ start of the gene, and a negative insertion site number indicates an insertion upstream of locus listed. Disrupted ECOR29 genes were mapped to strain MG1655, and, when appropriate, the MG1655 homolog locus tag and gene name are listed. Genes with ≥90% amino acid identity to a known *E. coli* gene have predicted gene names listed.
[b]Inferred from homology to MG1655 unless stated otherwise, see Table S1.
[c]Gene not found in MG1655, inferred from amino acid homology, see Table S1. Not applicable has been abbreviated as N/A here and throughout the table.
[d]Predicted to impact promoter of the indicated gene.

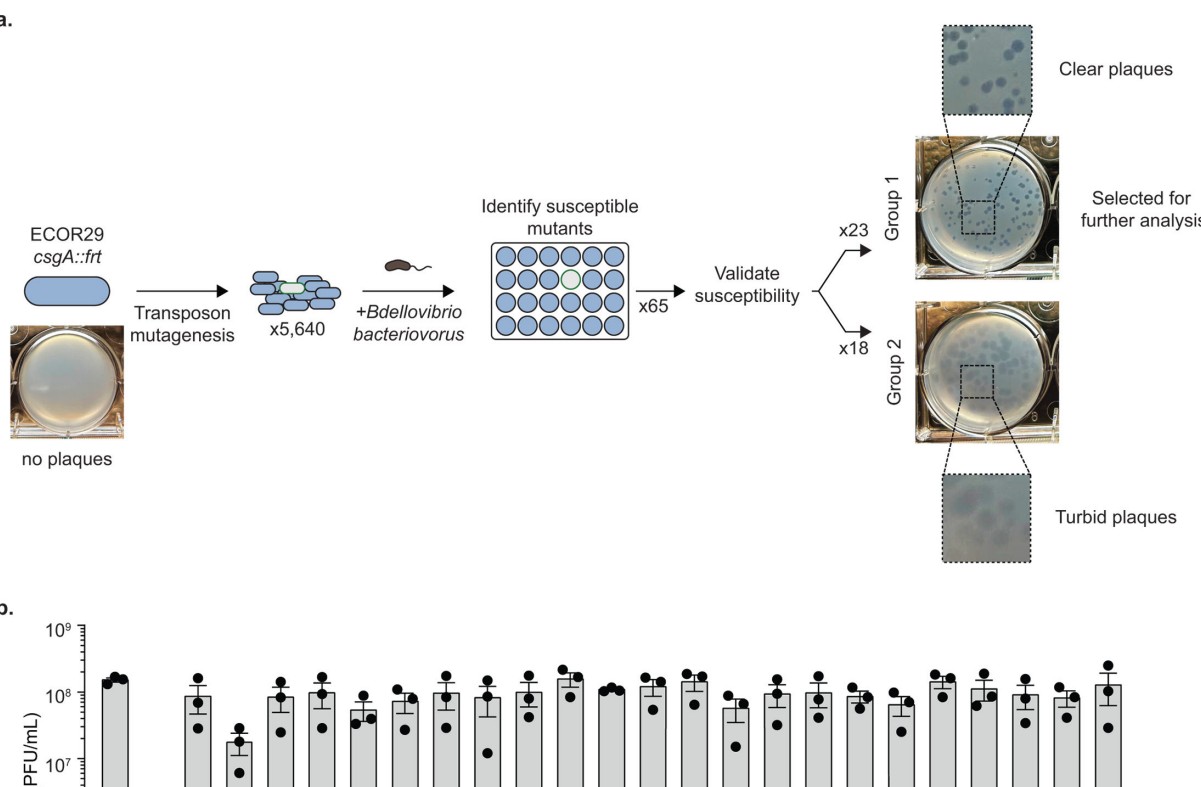

**FIG 2** A forward genetic screen for determinants of defense in ECOR29. (a) Schematic depicting the workflow of screening a transposon mutant library. Representative images of a single well from a 6-well dish double agar overlay experiment, infected at an MOI of approximately 0.001 are shown. (b) Efficiency of plating of *B. bacteriovorus* HD100 when infecting group 1 transposon mutants. Data are graphed as in Fig. 1a.

Deletion of *rpoS, pgi, fau, arnD, gspA, cra, waaV, lpp, wzzB, wzzE, ACS228_08725, rfaH,* and *rfe* resulted in loss of defense in both ECOR29 and ECOR29 *csgA::frt* (Fig. 3a and b). Many of these genes are involved in the production of crucial outer membrane components. For example, *waaV* encodes a glycosyl transferase involved in the assembly of the LPS outer core (20). Additionally, *lpp* encodes the lipoprotein Lpp, which regulates the distance between outer and inner membranes of *E. coli* (27). *wzzB* and *wzzE* encode polysaccharide co-polymerases that regulate oligosaccharide length and influence LPS and enterobacterial common antigen length, respectively (28). *ACS228_08725* is a hypothetical protein that is homologous to other O-antigen polymerases. *rfaH* is a transcriptional antiterminator that regulates expression of the *waa* operon (29), which is involved in the biosynthesis of LPS (20). Finally, *rfe* is the first step of *E. coli* enterobacterial common antigen and O-antigen synthesis (30). These genetic determinants indicate a critical role of the ECOR29 bacterial outer membrane in contributing to defense against *B. bacteriovorus*. The predicted function of all disrupted genes is detailed in Table S1.

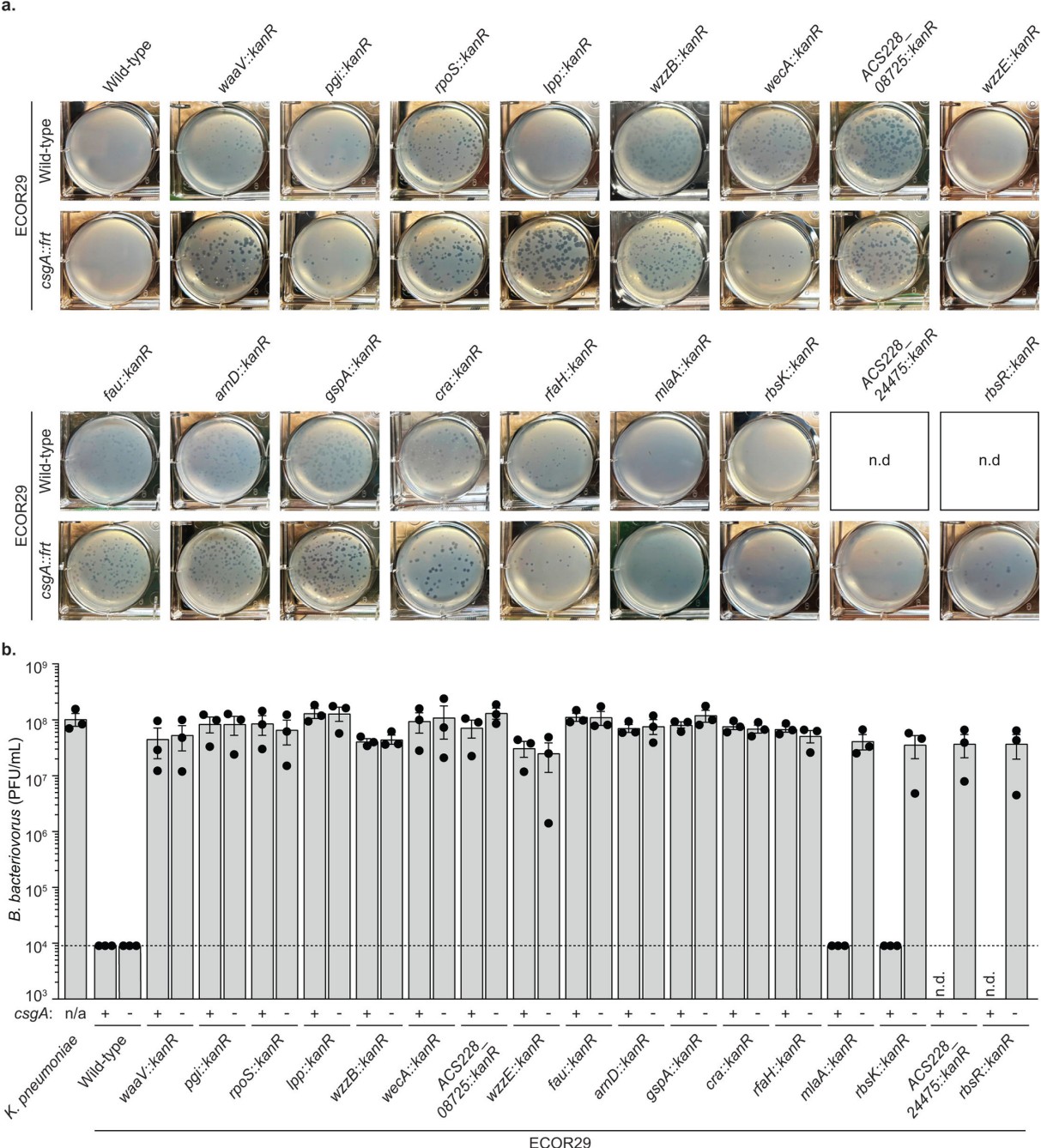

**FIG 3** ECOR29 mutations reveal genes that influence *B. bacteriovorus* defense. (a) Representative images depicting the plaque morphology of *B. bacteriovorus* when infecting the indicated bacterial strain. Images are of a single well of 6-well dish double-agar overlay infected at an MOI of approximately 0.001. Mutants unable to be constructed were not determined (n.d.). (b) Efficiency of plating of *B. bacteriovorus* HD100 when infecting the indicated bacterial strain. Data are graphed as in Fig. 1a. Mutants unable to be constructed were n.d.

A limitation of our transposon mutagenesis approach was the inability to interrogate how essential genes contribute to *B. bacteriovorus* resistance. For example, one of our mutations disrupted the intergenic region upstream of the essential gene *ispU*. Isoprenoid U catalyzes a step in the synthesis of undecaprenyl pyrophosphate, which functions as a lipid carrier for bacterial cell wall carbohydrates and is therefore essential (31). We did not attempt to construct a deletion of *ispU* and did not further interrogate this gene. However, identification of this transposon insertion suggests that proper

expression of *ispU* is important for defense. Additionally, it appeared that some identified transposon mutants may have additional mutations that we did not identify. Transposon mutants 808, 869, and 4543 harbored transposon insertions in *proC*, *tolC*, and *glpX*; however, deletion of these genes did not recapitulate a loss of *B. bacteriovorus* defense (Fig. S3).

In total, our results provide insights into the cellular processes that contribute to ECOR29 resistance to *B. bacteriovorus* predation.

## ECOR29 does not defend against *B. bacteriovorus* through an obvious extracellular structure, secreted factor, or a plasmid-encoded mechanism

The genetic determinants of ECOR29 *csgA::frt* resistance to *B. bacteriovorus* did not identify an obvious molecular mechanism; however, these mutations suggested three potential hypotheses: (i) ECOR29 produces an extracellular curli-like structure for defense; (ii) ECOR29 produces a secreted factor that antagonizes *B. bacteriovorus*; (iii) a plasmid resident to ECOR29 encodes for *B. bacteriovorus* defense. Any of these might require outer membrane integrity, thus explaining the phenotype for mutants recovered in our forward genetic screen. These explanations are not necessarily mutually exclusive.

Curli fibers are visible by transmission electron microscopy (17). To test whether ECOR29 employs a novel extracellular structure resembling curli, we used transmission electron microscopy to examine wild-type and *csgA* mutant ECOR29. Additionally, we included a strain lacking the LPS core modifier gene *waaV* alone or in combination with *csgA* to examine the role of outer membrane modifications in stabilizing alternative extracellular structures. No obvious external structures could be identified on the ECOR29 *csgA::frt* cell surface in the presence or absence of *waaV* (Fig. 4a). Production of curli fibers was less pronounced in ECOR29 compared to the strain ECOR14, which exclusively uses curli to defend against *B. bacteriovorus*. While no structures were visible, these data do not entirely rule out ECOR29 producing a novel extracellular structure because it is technically challenging to preserve extracellular matrices during preparation for microscopy (32).

We next sought to investigate if ECOR29 produces a secreted bactericidal factor that antagonizes *B. bacteriovorus*, similar to how cyanide production by *Chromobacterium piscinae* can inhibit *B. bacteriovorus* predation (15). *B. bacteriovorus* was challenged with spent medium from ECOR29 in the presence or absence of *csgA*. We also incubated with spent supernatant from ECOR29 *csgA::frt; mlaA::kanR*. We selected MlaA because this protein was implicated in virulence of *E. coli* in a silkworm infection model, and we hypothesized MlaA might have a large impact on many secreted factors (33). Additionally, we included supernatants from cultures of *K. pneumoniae*, ECOR14, and a media-only condition as negative controls. None of the filter-sterilized supernatants impacted *B. bacteriovorus* viability, as measured by plaque formation on *K. pneumoniae*, the indicator strain, even when incubation spanned 48 hours (Fig. 4b). These data show that under conditions showing defense, ECOR29 does not secrete a factor that impacts *B. bacteriovorus* viability or predation. Importantly, however, we cannot rule out a hypothetical secreted factor with an extremely short half-life.

Finally, our screen identified that the mutation of *ACS228_24475*, a gene on a plasmid in ECOR29 encoding a protein predicted to be involved in plasmid partitioning (26), confered resistance to *B. bacteriovorus*. Genes for defense against threats such as bacteriophages are commonly located in mobile genetic elements within so-called "defense islands" (34–37). We therefore investigated the role of the ECOR29 plasmid (pECOR29, the only plasmid identified in this strain) in *B. bacteriovorus* defense and hypothesized that ECOR29 defense was encoded on pECOR29. We conjugated a marked version of pECOR29 into the susceptible strain ECOR14 *csgA::frt* and found that pECOR29 was not sufficient to provide defense (Fig. 4c). An ideal experiment would be to cure ECOR29 of pECOR29 and examine if the resulting strain was no longer resistant to *B. bacteriovorus*. Unfortunately, pECOR29 encodes multiple candidate addiction modules, and we were unable to isolate ECOR29 mutants that had lost pECOR29. These data

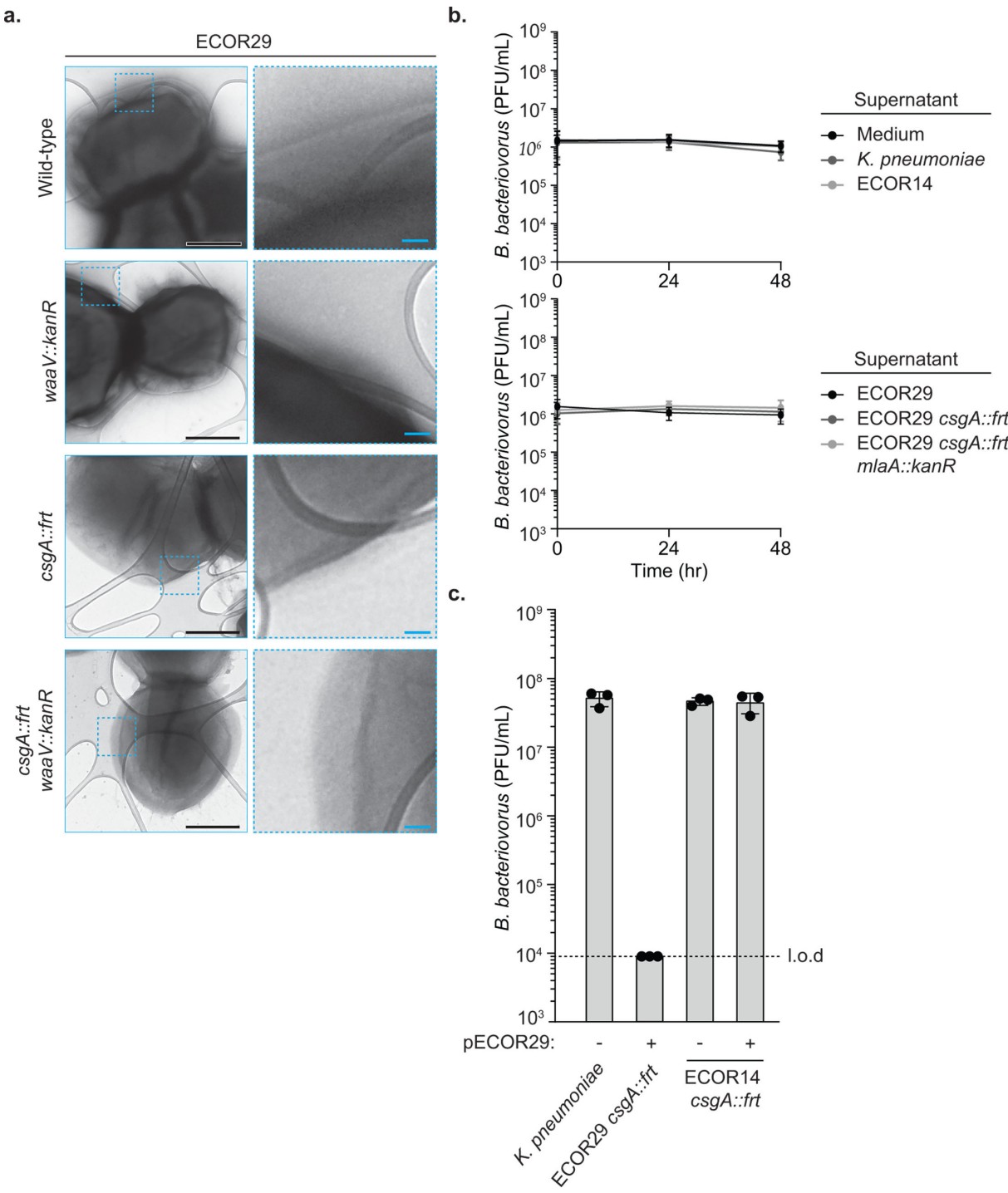

**FIG 4** Investigating the role of external structures, secreted factors, and mobile genetic elements in ECOR29 defense. (a) Representative transmission electron microscopy images of indicated bacterial strain alongside zoom-in on area depicted by the blue dotted-line box to examine outer membrane features. Black scale bars are 500 nm, and blue scale bars are 50 nm. (b) Efficiency of plating of *B. bacteriovorus* HD100 incubated with indicated supernatant for the indicated amount of time. Data are the mean ± standard error of the mean for *n* = 3 biological replicates. (c) Efficiency of plating of *B. bacteriovorus* HD100 when infecting the indicated bacterial strain. Data are graphed as in Fig. 1a.

showed that it is unlikely pECOR29 encodes a novel *B. bacteriovorus* defense system; however, we cannot rule out that pECOR29 and its hypothetical defense system simply requires a specialized host factor missing in ECOR14.

Our genetic screen identified many intrinsic features such as outer membrane modifying enzymes in ECOR29 that mediated resistance to *B. bacteriovorus*. Our findings led us to investigate the role of extracellular structures, secreted factors, and mobile genetic elements in ECOR29 curli-independent defense, and we found no evidence of a role of these factors in *B. bacteriovorus* defense.

## DISCUSSION

In this study, we identified and investigated a curli-independent mechanism for *B. bacteriovorus* resistance and found genetic determinants associated with defense. Previously, our lab surveyed a collection of *E. coli* strains and uncovered numerous strains that naturally resist predation. Production of curli fibers was the dominant defense mechanism. Here, we identify a single strain, ECOR29, that employs a unique curli-independent mechanism to restrict predation. Transposon mutagenesis of ECOR29 led us to investigate the roles of extracellular structures, secreted factors, and mobile genetic elements. Our findings suggest that ECOR29 defense relies on a properly regulated outer membrane. Unfortunately, the exact mechanism for defense remains unidentified and is challenging to pinpoint because the identified genetic determinants are involved in general aspects of outer membrane maintenance or otherwise failed to coalesce into a unified theme. A future direction for this work will be investigating the proteome of isolated ECOR29 mutants and/or continued genetic screens (our screen failed to reach saturation as evidenced by only rarely identifying transposon insertions in the same genes).

The ECOR29 genetic determinants of defense clearly implicate central cell processes at the cell envelope as crucial for defense, including outer membrane maintenance, LPS modifications, and outer membrane polysaccharides. ECOR29 shares an O150 serotype with ECOR22 (20), which is susceptible to *B. bacteriovorus* infection (17). This suggests that O-antigen likely does not specifically confer defense against *B. bacteriovorus* predation. However, our findings for the genetic determinants of ECOR29 resistance appear similar to a contemporary study that experimentally evolved strains of K-12 *E. coli* to resist predation, which identified that *ompF* and *waaF* mutations lead to an increase in resistance (16). Taken together with previous studies showing OmpF is important for *B. bacteriovorus* entry (38) and that different LPS outer cores (39) and O-antigen modifications alter predator attachment (6), ECOR29 defense further demonstrates that a complicated set of interactions at the cell envelope modulates susceptibility or resistance to *B. bacteriovorus*.

## MATERIALS AND METHODS

### Bacterial strains and growth conditions

Bacterial strains used in this study are listed in Table S2. *E. coli* and *K. pneumoniae* strains were cultured in LB medium (1% tryptone, 0.5% yeast extract, 0.5% NaCl) or YT medium (0.8% tryptone, 0.5% Bacto Peptone, 0.5% NaCl) shaking at 37°C, 220 rpm, unless otherwise noted. We defined an "overnight" as a culture grown 16–20 hours postinoculation from a single colony or a glycerol stock. Strains are stored in LB plus 30% glycerol and kept at −70°C.

### *Bdellovibrio bacteriovorus* amplification and storage

*B. bacteriovorus* was initially revived from a frozen stock using a double-agar overlay as previously described (17). *Klebsiella pneumoniae* was grown overnight in YT medium at 37°C, 220 rpm. Furthermore, 400 µL of overnight was then mixed with 3.5 mL YPSC soft agar (0.125 g/L MgSO$_4$, 3 mM NaOAc, 0.5 g/L Bacto Peptone, 0.5 g/L yeast extract, 2.3 mM CaCl$_2$, and 0.6% agar) and poured onto a YPSC plate (0.125 g/L MgSO$_4$, 3 mM NaOAc, 0.5 g/L Bacto Peptone, 0.5 g/L yeast extract, 2.3 mM CaCl$_2$, and 1% agar). The top agar was left at room temperature for 10 min to solidify, and plates were incubated

overnight at 30°C. The following day, ~100 µL of freezer stock (*B. bacteriovorus* lysate in 30% glycerol) was spotted onto the soft agar mixture with *K. pneumoniae* and incubated at 30°C for 3–5 days till visible clearance was observed. The zone of clearance was then extracted using a sterile razor blade, and the agar plug was added to a flask containing *K. pneumoniae* diluted 1:20 in diluted nutrient broth (DNB; 0.1% tryptone, 0.05% yeast extract, 0.05% NaCl, 3 mM MgCl$_2$, and 2 mM CaCl$_2$) from an overnight culture in YT. The mixture was left to incubate at 30°C, 220 rpm till culture became clear (2–3 days). The working lysate was then extracted using centrifugation (10 min at 4,000 × *g*) and filtered with a 0.45 µm Nanosep filter.

Working *B. bacteriovorus* lysates were generated every 7 days via liquid propagation of revived stock. Overnight cultures of *K. pneumoniae* grown in YT medium were diluted 1:10 in 20 mL DNB and left overnight at 30°C, 220 rpm. The next day, cultures were infected at an MOI of ~0.01 and left shaking for 3 days until the culture became clear. Working lysates were then collected as described in "Bacterial strains and growth conditions." Titers were determined by double-agar overlay assays (described in "*B. bacteriovorus* enumeration") against *K. pneumoniae*. *B. bacteriovorus* working lysates were stored at 4°C in DNB. Freezer stocks were generated by mixing freshly generated lysate with 60% glycerol to final concentration of 30% glycerol and stored at −70°C.

## Efficiency of plating/*B. bacteriovorus* predation assays

*B. bacteriovorus* predation assays were performed as described previously (17). Prey strains were cultivated overnight in YT medium at 37°C, 220 rpm. In standard predation assays, we used 6-well dishes to plate serial fold dilutions of *B. bacteriovorus* against the same concentration of prey. Furthermore, 200 µL of prey strain plus 100 µL of *B. bacteriovorus* dilutions ($10^0$ through $10^{-5}$ was used in each experiment) was added to 750 µL of 0.6% YPSC agar, mixed, and poured onto 6-well dishes containing YPSC 1% agar. Plates were left at room temperature for 10 min to solidify and then inverted and incubated at 30°C for 3 days. Plates were imaged after 3 days. PFU/mL were enumerated, and the resulting efficiency of plating was measured.

## *B. bacteriovorus* enumeration

Using our standard *B. bacteriovorus* predation assays, quantifiable plaques are only observed in $10^{-3}$ through $10^{-5}$ dilutions of *B. bacteriovorus*. Our assays show that some mutations in ECOR29 yielded turbid plaques. More turbid plaques were difficult to see and are hypothesized to reflect a mild restriction in *B. bacteriovorus* predation. When no quantifiable plaques were observed, 0.9 PFU at $10^{-3}$ was used as the limit of detection for that assay.

## Transposon mutagenesis library generation

We used a mariner transposon to generate loss-of-function mutants in ECOR29 as previously described (17, 40). *E. coli* MFD (41) transformed with pSC189 (42) was used as a donor strain. The recipient, ECOR29 *csgA::frt*, and donor strain were grown overnight, and 1 mL of each strain was pelleted via centrifugation, washed in 1 mL of LB supplemented with 300 µM DAP (2,6-diaminopimelic acid), and again pelleted via centrifugation. Both recipient and donor were combined by resuspending both in the same 50 µL of LB plus 300 µM DAP. This mixed mating condition was spotted onto a mating filter (Pall Corporation) placed on an LB agar plate supplemented with 300 µM DAP. Spots were left to dry for 10 min at room temperature, and the plate was incubated at 37°C for 1 hour. After incubation, the mating spot was resuspended in a 15 mL conical tube with 2 mL of LB medium via a light vortex for 1 min. Loss-of-function mutants were kanamycin resistant and were selected for by plating 300 µL onto 6 × 15 cm$^2$ LB plates supplemented with 25 µg/mL kanamycin. Three plates of $10^0$ and $10^{-1}$ dilutions of the resuspended conjugate were plated to ensure single colonies form.

Individual colonies were picked and then gridded onto a 96-well plate containing 200 µL LB plus 25 µg/mL kanamycin and grown overnight. Each 96-well plate contained

two wells dedicated to controls, one containing *K. pneumoniae* (susceptible control) and one containing ECOR29 *csgA::frt* (resistant control), which were inoculated in LB medium not supplemented with antibiotics. The next day, glycerol was added to a final concentration of 10%, and libraries were stored at −70°C. Mutagenesis library generation was conducted three separate times for 1,880 mutants per library (total of 5,640 mutants).

## Transposon mutagenesis library screening

We used a high-throughput method to screen Tn mutants as previously described (17). Briefly, we used a 24-well-based predation assay that is a modification of the above described 6-well assay. Mutants and controls were inoculated from frozen stocks in a 96-well plate with 200 µL of YT medium and grown overnight at 37°C with 220 rpm. In a fresh 96-well plate, 50 µL of prey was mixed with 100 µL of *B. bacteriovorus* with a final MOI = 0.01. This mixture was added to 750 µL 0.6% YPSC top agar in a deep 96-well plate in a heat block kept at 60°C. Using an adjustable 6-well multichannel pipet, 750 µL of YPSC agar containing the bacterial mixture was transferred to 24-well plates containing 500 µL 1.0% YPSC agar. The plate was left to solidify for ~10 min at room temperature and then incubated inverted at 30°C for 3 days. Wells were checked daily. On day 1, strains that did not grow overnight were noted and removed from this study. At days 2 and 3, any well displaying clearance or reduced opacity was noted as hits.

## ECOR29 genome sequencing

ECOR29 genomic DNA was extracted using the Qiagen DNeasy Blood & Tissue Kit. The gDNA samples were submitted for Illumina and Nanopore hybrid sequencing performed by SeqCenter (Pittsburgh, PA). Briefly, sample libraries were prepared using the Illumina DNA Prep Kit and IDT 10 bp UDI indices, and sequenced on an Illumina NextSeq 2000, producing 2 × 151 bp reads. Demultiplexing, quality control, and adapter trimming were performed with bcl-convert (v3.9.3). The samples also underwent Oxford Nanopore Technology Ligation Sequencing library preparation with a native barcode kit providing a median raw read accuracy of Q20+. The ECOR29 genome sequence was deposited in GenBank at the National Center for Biotechnology Information/National Library of Medicine/National Institutes of Health under: bioproject PRJNA1298331; biosample SAMN50266478; accessions CP197071 (chromosome) and CP197072 (plasmid, pECOR29).

## Arbitrarily primed PCR transposon mapping

Transposon mutants identified as hits from above were mapped using arbitrary PCR as previously described (43, 44). Briefly, genomic DNA isolated from the Qiagen DNeasy Blood & Tissue Kit or an inoculate of mutant glycerol stock was used as a template for the first round of PCR with primers Himar 1.2 (GGGCTGATCGCTTCCTCGTGCTTTAC) and Arb 1 (GGCCACGCGTCGACTAGTACNNNNNNNNNNNCTTCT) (primers listed in Table S2). Furthermore, 1 µL of product was used as a template for the second round of PCR amplified with Himar 2.2 (GGTATCGCCGCTCCCGATTCGCAGC) and Arb 2 (GGCCACG CGTCGACTAGTAC). The reaction was then purified with Exo-CIP (New England Biolabs) treatment per manufacturer's instructions. PCR products were then Sanger sequenced (Azenta or Quintara Biosciences) using the Himar 3.3 sequencing primer (CCTATTCTCTA GAAAGTATAGGAACTTCGAACC). The resulting sequence was aligned to the genome of ECOR29 (accessions CP197071 and CP197072), and the "CCTGTTA" site at the 5′ end of the transposon was found to identify the precise TA insertion site in disrupted loci. BLAST (45), HHpred (46, 47), conserved domain database (48), and InterPro (49) predictions were used to determine gene identity and predicted function.

## Constructing marked deletions in *E. coli*

We employed lambda red recombineering to generate marked deletions as previously described (17, 50, 51). Briefly, a kanamycin resistance cassette including *frt* sites was

amplified from pKD4 with 50 bp of homologous regions flanking target genes that preserved the first and last six amino acids and a stop codon. The product was PCR purified. The target strains (ECOR29 and ECOR29 *csgA::frt*) were transformed with pKD46 through electroporation and recovered at 30°C in LB medium supplemented with 100 µg/mL of carbenicillin and 0.2% glucose.

To generate mutants, target strains were grown overnight at 30°C in LB medium supplemented with 100 µg/mL of carbenicillin and 0.2% glucose. The next day, cells were back diluted 1:100 in 25 mL of LB without salts plus 100 µg/mL of carbenicillin and 0.2% arabinose to induce lambda red machinery. Cells were grown to an $OD_{600} = 0.4$–$0.6$, pelleted, washed three times in cold sterile water, and resuspended in 200 µL of sterile water. And, 50 µL of electrocompetent cells was transformed with 5 µL of purified PCR product via electroporation and was recovered in LB plus 0.2% arabinose for 2 hours at 30°C and then plated on LB agar supplemented with 25 µg/mL kanamycin at 30°C overnight.

The next day, colonies were patched onto LB agar plus 50 µg/mL kanamycin (to look for proper integration of kanamycin cassette) and LB agar plus 50 µg/mL carbenicillin (to screen for loss of temperature-sensitive pKD46). Patch plates were grown overnight at 37°C, and clones grown on kanamycin, but not carbenicillin, were then PCR screened using SapphireAmp 2× Master Mix (Takara) to look for proper integration of kanamycin cassette (primers listed in Table S3). Positive clones were selected and used for downstream experiments.

The strains ECOR10 and ECOR31 were already resistant to carbenicillin, consistent with previous studies (52), and thus we were unable to transform each with pKD46. No carbenicillin-resistant colonies could be recovered when ECOR58 was transformed with pKD46. ECOR28 could be transformed with pKD46, but we were not able to obtain kanamycin-resistant colonies.

## Excising kanamycin cassette from marked deletion mutations

We used FLP recombinase to excise the kanamycin resistance cassette as previously described (53). Briefly, we transformed our marked deletion mutant with pCP20 using electroporation. Transformants recovered in a rich medium for 2 hours at 30°C at 220 rpm. Furthermore, 50 µL of $10^0$ and $10^{-1}$ dilutions of recovery was plated onto LB plus 100 µg/mL carbenicillin and incubated overnight at 30°C. The next day, three colonies were picked, inoculated into 5 mL of LB medium, and grown at 42°C at 220 rpm to induce FLP recombinase expression. The next day, 100 µL of $10^0$–$10^{-8}$ dilutions was plated onto LB and incubated overnight at 30°C. Then, clones were picked from the highest dilution, where individual colonies were distinguishable and then patched onto LB + 50 µg/mL of kanamycin (to look for loss of resistance), LB + 100 µg/mL carbenicillin (to look for loss of pCP20), and LB-only plates (to isolate successful clone). All patch plates were incubated at 37°C except for LB + 100 µg/mL carbenicillin, which was incubated at 30°C. The next day, clones that grew only on LB were inoculated into 5 mL of LB medium and grown overnight at 37°C shaking. The next day, cultures were mixed with 60% glycerol to a final concentration of 30% and stored at −70°C. For validation, generated strains were PCR screened using SapphireAmp 2× Master Mix (Takara) and the respective primers used in lambda red recombineering to validate proper loss of kanamycin cassette.

## Transmission electron microscopy

We prepared *E. coli* strains for transmission electron microscopy as previously described (17). Briefly, strains were grown overnight in YT medium at 37°C with 220 rpm. The next day, cells were diluted 1:10 into 20 mL of DNB and grown for 24 hours at 30°C with 220 rpm. The next day, cells were concentrated 100-fold via centrifugation to obtain ~$10^{11}$ CFU/mL. Furthermore, 5 µL of each concentrated sample was spotted onto a 300-mesh lacey carbon film with a continuous layer of ultrathin carbon film (Electron

Microscopy Sciences). Samples were then stained with 2% uranyl acetate and imaged. Images were edited to improve visualization; unedited images are shown in Fig. S4.

## Treating *B. bacteriovorus* with prey supernatant

To mimic predation conditions, strains were grown overnight in 5 mL of DNB at 30°C with 220 rpm. The next day, 1 mL of *B. bacteriovorus* HD100 was inoculated into 20 mL of DNB, and the prey overnight was pelleted using centrifugation, and supernatant was filter sterilized using a 0.22 µm Nanosep filter and added to HD100 + DNB mixture. *B. bacteriovorus* PFU/mL were enumerated using the 6-well double-agar overlay described above using *K. pneumoniae* as a susceptible strain. PFU/mL were collected at 0, 24, and 48 hours post incubation with prey supernatant or DNB-only medium.

## Testing sufficiency of pECOR29

We generated a marked deletion in the gene *ACS228_15805* on the plasmid harbored by ECOR29 (pECOR29). *ACS228_15805* is homologous to *relE* (54) and had no impact on ECOR29 susceptibility to *B. bacteriovorus* predation (Fig. S2). We then transformed ECOR29 *ACS228_15805::kanR* with a plasmid expressing sfGFP (55) (see Table S2) using electroporation. ECOR29 *ACS228_15805::kanR* + pTACxc-sfGFP was used as a donor strain to conjugate into ECOR14 *csgA::frt*. Donor and recipient were grown at 37°C at 220 rpm in LB + 50 µg/mL kanamycin + 100 µg/mL carbenicillin and LB only, respectively. Donor and recipient were pelleted and mixed with LB medium as described in "Transposon mutagenesis library generation methods." Dilutions of $10^0$–$10^{-3}$ were plated on LB plus 50 µg/mL of kanamycin and incubated at 37°C overnight. Individual clones were patched on LB + 50 µg/mL kanamycin (to select for positive conjugates of ECOR14 *csgA::frt* + pECOR29) and on LB + 100 µg/mL carbenicillin (positive conjugates should be carbenicillin sensitive). Kanamycin-resistant and carbenicillin-sensitive strains were used for further experiments.

## ACKNOWLEDGMENTS

The authors would like to thank Daniel Kadouri (Rutgers University) for sharing bacterial strains; the CU Boulder Department of Biochemistry Shared Instruments Pool core facility (RRID:SCR_018986), Annette Erbse, and its staff; Garry Morgan and Sarah Ann Zimmerman in the Boulder Electron Microscopy Services Core Facility (RRID:SCR_001432) for electron microscopy sample preparation and imaging; and members of the Whiteley lab for their insight and helpful discussion.

This work was funded by the National Institutes of Health through the NIH Director's New Innovator Award DP2AT012346 and the NIH Common Fund 3DP2AT012346-01S1 (A.T.W.), and the Boettcher Foundation's Webb-Waring Biomedical Research Program (A.T.W.). R.S. was supported in part by the NIH T32 Signaling and Cellular Regulation training grant (T32GM142607); H.E.L. was supported in part as a fellow of the Jane Coffin Childs Memorial Fund for Medical Research.

Experiments were designed and conceived by R.S. and A.T.W. Marked deletions in Fig. 1 were made by R.S. and H.E.L. All other experiments were performed by R.S. Figures were prepared by R.S. and A.T.W. The manuscript was written by R.S. and A.T.W. All authors contributed to editing the manuscript and supported the conclusions.

## AUTHOR AFFILIATIONS

[1]Department of Biochemistry, University of Colorado Boulder, Boulder, Colorado, USA
[2]Department of Molecular, Cellular and Developmental Biology, University of Colorado Boulder, Boulder, Colorado, USA

## AUTHOR ORCIDs

Aaron T. Whiteley ⓘ http://orcid.org/0000-0002-0075-7519

## FUNDING

| Funder | Grant(s) | Author(s) |
|---|---|---|
| National Institutes of Health | DP2AT012346 | Aaron T. Whiteley |
| Boettcher Foundation | | Aaron T. Whiteley |
| National Institutes of Health | T32GM142607 | Ryan Sayegh |
| National Institutes of Health | 3DP2AT012346-01S1 | Aaron T. Whiteley |
| Jane Coffin Childs Memorial Fund for Medical Research | | Hannah E. Ledvina |

## AUTHOR CONTRIBUTIONS

Ryan Sayegh, Conceptualization, Data curation, Formal analysis, Investigation, Methodology, Visualization, Writing – original draft, Writing – review and editing | Hannah E. Ledvina, Investigation, Methodology, Resources, Writing – review and editing | Aaron T. Whiteley, Conceptualization, Project administration, Supervision, Visualization, Writing – original draft, Writing – review and editing

## ADDITIONAL FILES

The following material is available online.

### Supplemental Material

**Figures S1 to S4 (Spectrum00342-25-s0001.pdf).** Fig. S1: Group 2 transposon mutants *B. bacteriovorus* susceptibility. Fig. S2: Marked deletions in lacZ and ACS228_15805 do not impact ECOR29 defense. Fig. S3. Deletions in Group 1 genes proC, tolC, and glpX did not impact ECOR29. Fig. S4: Unedited transmission electron microscopy images.
**Table S1 (Spectrum00342-25-s0002.xlsx).** Group 1 and Group 2 transposon mutants.
**Table S2 (Spectrum00342-25-s0003.xlsx).** Bacterial strains used in this study.
**Table S3 (Spectrum00342-25-s0004.xlsx).** Oligonucleotides used in this study.

### Open Peer Review

**PEER REVIEW HISTORY (review-history.pdf).** An accounting of the reviewer comments and feedback.

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
