## [Reviewer comments · Microbiology Spectrum]

Microbiology Spectrum

Curli-independent defense against *Bdellovibrio bacteriovorus* in *E. coli*

Ryan Sayegh, Hannah Ledvina, and Aaron Whiteley

Corresponding Author(s): Aaron Whiteley, University of Colorado Boulder

Review Timeline:

Submission Date:	March 21, 2025
Editorial Decision:	June 4, 2025
Revision Received:	August 17, 2025
Accepted:	August 29, 2025

Editor: Cheryl Andam

Reviewer(s): The reviewers have opted to remain anonymous.

Transaction Report:

DOI: <https://doi.org/10.1128/spectrum.00342-25>

Re: Spectrum00342-25 (Curli-independent defense against *Bdellovibrio bacteriovorus* in *E. coli*)

Dear Dr. Aaron Whiteley:

Thank you for the privilege of reviewing your work. Below you will find my comments, instructions from the Spectrum editorial office, and the reviewer comments.

The paper is editorially accepted. Please answer the additional questions on the submission form related to funding and publication costs. When the revision is received, I will send the accept decision.

Revision Guidelines

Sincerely,
Eric Cascales
Editor
Microbiology Spectrum

Reviewer #1 (Comments for the Author):

There is lots to like about the work. The authors have thoughtfully responded to my comments, but I just don't know if this rises to the level of an mSphere manuscript without some better precision in determining why ECOR29 displays curli-independent resistance to *B. bacteriovorus*. However, the work is well done and should be published.

Response to Reviewers

Below, we have responded to each of the reviewer comments in blue text.

Reviewer 1:

There is lots to like about the work. The authors have thoughtfully responded to my comments, but I just don't know if this rises to the level of an mSphere manuscript without some better precision in determining why ECOR29 displays curli-independent resistance to B. bacteriovorus. However, the work is well done and should be published.

We thank the reviewer for their critiques.

Re: Spectrum00342-25R1 (Curli-independent defense against *Bdellovibrio bacteriovorus* in *E. coli*)

Dear Dr. Aaron Whiteley:

Your manuscript has been accepted, and I am forwarding it to the ASM production staff for publication. Your paper will first be checked to make sure all elements meet the technical requirements. ASM staff will contact you if anything needs to be revised before copyediting and production can begin. Otherwise, you will be notified when your proofs are ready to be viewed.

Sincerely,
Cheryl Andam
Editor
Microbiology Spectrum